# Evaluation of different infant vaccination schedules incorporating pneumococcal vaccination (The Vietnam Pneumococcal Project): protocol of a randomised controlled trial

Beth Temple,[1,2,3] Nguyen Trong Toan,[4] Doan Y Uyen,[4] Anne Balloch,[3] Kathryn Bright,[3] Yin Bun Cheung,[5,6] Paul Licciardi,[3,7] Cattram Duong Nguyen,[3,7] Nguyen Thi Minh Phuong,[4] Catherine Satzke,[3,7] Heidi Smith-Vaughan,[8] Thi Que Huong Vu,[9] Tran Ngoc Huu,[4] Edward Kim Mulholland[2,3]

TQHV is deceased.

## ABSTRACT

**Introduction** WHO recommends the use of pneumococcal conjugate vaccine (PCV) as a priority. However, there are many countries yet to introduce PCV, especially in Asia. This trial aims to evaluate different PCV schedules and to provide a head-to-head comparison of PCV10 and PCV13 in order to generate evidence to assist with decisions regarding PCV introduction. Schedules will be compared in relation to their immunogenicity and impact on nasopharyngeal carriage of *Streptococcus pneumoniae* and *Haemophilus influenzae*.

**Methods and analysis** This randomised, single-blind controlled trial involves 1200 infants recruited at 2 months of age to one of six infant PCV schedules: PCV10 in a 3+1, 3+0, 2+1 or two-dose schedule; PCV13 in a 2+1 schedule; and controls that receive two doses of PCV10 and 18 and 24 months. An additional control group of 200 children is recruited at 18 months that receive one dose of PCV10 at 24 months. All participants are followed up until 24 months of age. The primary outcome is the post-primary series immunogenicity, expressed as the proportions of participants with serotype-specific antibody levels ≥0.35 µg/mL for each serotype in PCV10.

**Ethics and dissemination** Ethical approval has been obtained from the Human Research Ethics Committee of the Northern Territory Department of Health and Menzies School of Health Research (EC00153) and the Vietnam Ministry of Health Ethics Committee. The results, interpretation and conclusions will be presented to parents and guardians, at national and international conferences, and published in peer-reviewed open access journals.

**Trial registration number** NCT01953510; Pre-results.

## Strengths and limitations of this study

► This study is specifically designed to address two independent questions within a single study: which schedule to use for the provision of pneumococcal conjugate vaccine (PCV), and which PCV to use.
► This study includes a head-to-head comparison of the two licensed PCVs, allowing a direct assessment of their relative immunogenicity and impact on nasopharyngeal carriage.
► The primary outcome is the criteria used for the licensing and varying of PCV schedules.
► This study has relatively low power for the secondary nasopharyngeal carriage outcomes, so the ability to draw conclusions relating to these outcomes is vulnerable in the event of lower-than-anticipated carriage rates.

pneumococcal conjugate vaccine (PCV7), was licensed in the USA in the year 2000. Introduction of PCV7 has been associated with dramatic reductions in pneumococcal disease.[1–3] However, geographical variation in serotype distribution[4–7] and an increase in invasive pneumococcal disease (IPD) caused by non-PCV7 serotypes following vaccine introduction[8] necessitated the development of higher valency PCVs.

There are currently two licensed PCVs: PCV10, a 10-valent pneumococcal vaccine that uses non-typeable *Haemophilus influenzae* (NTHi) protein D as a carrier protein for 8 of the 10 serotypes (*Synflorix*, PHiD-CV; GSK); and PCV13, a 13-valent pneumococcal $CRM_{197}$ conjugate vaccine (*Prevnar-13/Prevenar-13*; Pfizer). Both have been shown to be non-inferior to PCV7 in terms of post-primary series immunogenicity for the shared serotypes.[9–11]

For numbered affiliations see end of article.

**Correspondence to**
Beth Temple;
beth.temple@menzies.edu.au

## INTRODUCTION
### Background and rationale

*Streptococcus pneumoniae* (pneumococcus) remains a leading vaccine preventable cause of serious infection in young children, despite the availability of effective vaccines. The first infant pneumococcal vaccine, the 7-valent

Despite the availability of both PCV10 and PCV13 for several years, there have been no published studies to date directly comparing their post-primary series immunogenicity or impact on nasopharyngeal (NP) carriage.

The cost of PCVs is a major barrier to vaccine introduction in low-income to middle-income countries; therefore, investigation of alternative schedules with a reduced number of doses is of great importance. The uptake of PCV introduction in Asia has been particularly slow. Three schedules are currently in routine use around the world for PCV introduction: a 3+1 schedule (a three-dose primary series followed by a booster dose in the second year of life), a 3+0 schedule (a three-dose primary series without a booster dose) and a 2+1 schedule (a two-dose primary series followed by a booster dose in the second year of life). Data from periods of PCV7 shortage in the USA show high vaccine effectiveness of a two-dose primary series against IPD,[12] [13] and trial data of CRM$_{197}$-conjugated PCVs show comparable immunogenicity following a two-dose or three-dose primary series, although antibody levels to serotypes 6B and 23F tend to be lower after two doses.[14] [15] Trials of PCV10 and PCV13 also support the use of a two-dose primary series. A trial of PCV10 in Europe directly comparing the immunogenicity of a two-dose and three-dose primary series showed a similar proportion of participants achieving protective antibody levels (≥0.2 µg/mL) for all 10 serotypes.[16] In a trial of PCV13 in Mexico, over 93% of participants achieved protective antibody levels (≥0.35 µg/mL) for most of the 13 serotypes following two doses, with the exception of serotypes 6B and 23F.[17] Four trials in Europe directly comparing PCV13 and PCV7 responses showed comparable immune responses between the vaccines following two doses.[18]

In developing countries, a 2+1 schedule with a booster dose in the first year of life may be advantageous. This modified schedule would likely increase compliance, would provide full immunisation closer to the peak incidence of pneumococcal disease and could enable the booster dose to coincide with measles vaccination. Alternatively, a further reduced PCV schedule with only two doses may be optimal for pneumococcal vaccination. Our previous trial in Fiji showed that protective antibody levels were reached for five of the seven serotypes following a single dose of PCV7 at 14 weeks of age.[15] Furthermore, a booster dose of the 23-valent pneumococcal polysaccharide vaccine at 12 months of age was more immunogenic following a single dose primary series of PCV7 compared with a two-dose or three-dose primary series for four serotypes, and comparable for the other three serotypes.[19] A trial of PCV9 from South Africa also showed that one dose at 6 weeks of age elicited a significant response for seven serotypes,[20] and modelling data from the USA suggest that a single dose of PCV could prevent up to 62% of IPD.[21] More recently, in the UK, where routine infant PCV vaccination has been in place for over 10 years, a 1+1 schedule of PCV13 was shown to elicit equivalent or superior post-booster responses compared with a 2+1 schedule for nine serotypes.[22]

Carriage of pneumococci in the nasopharynx is commonly a prerequisite for IPD and is the usual means of transmission of the bacteria. The herd effect of pneumococcal vaccination is mediated by the impact on NP carriage.[23] Vaccination with PCVs generally results in a decrease in vaccine type (VT) pneumococcal carriage, which is most commonly observed after a booster dose and often accompanied by a compensatory increase in non-VT carriage.[23–27] There have been few trials that evaluate the effect of different PCV schedules on carriage. A trial from the Netherlands showed that a two-dose primary series with or without a booster reduced VT carriage at 12 months of age compared with controls.[28] VT carriage was further reduced at 18 months in the group that received the booster dose, compared with the group that did not receive the booster, although this difference did not persist at 24 months of age. Similarly, our trial in Fiji showed that a two-dose or three-dose primary series with or without a booster reduced VT carriage at 12 months of age compared with controls, but no difference was seen at 17 months of age (F Russell, personal communication).

It has been hypothesised that the protein D carrier in PCV10 may result in an impact on *H. influenzae* carriage. A recent review of the impact of protein D-containing PCVs on NTHi carriage concludes that any such impact is likely to be small and transient, although changes in the density of carriage are yet to be evaluated. Two large phase III trials (POET trial of an 11-valent PCV and COMPAS trial of PCV10) showed trends towards a reduction in NTHi carriage following a booster dose of PCV, along with a trial of PCV10 in toddlers in Kenya, but other trials conducted in Finland, the Netherlands and the Czech Republic showed no impact of PCV10 on NTHi carriage.[29]

This trial includes six infant vaccination schedules: four different PCV10 schedules (arm A, a 3+1 schedule at 2, 3, 4 and 9 months of age; arm B, a 3+0 schedule at 2, 3 and 4 months; arm C, a 2+1 schedule at 2, 4 and 9.5 months; and arm D, a two-dose schedule at 2 and 6 months); a 2+1 PCV13 schedule at 2, 4 and 9.5 months (arm E); and a control group that receives two doses of PCV10 at 18 and 24 months (arm F). In response to more recent interest in schedules with only one or two doses of PCV, which may be sufficient to maintain herd immunity at the population level, an additional control group is recruited at 18 months of age for comparison with the initial control group (arm G).

### Explanation for choice of comparators

There was no PCV licensed in Vietnam at the time the protocol was finalised in 2013. The inclusion of control groups that receive no infant doses of PCV is therefore justified. Control group participants recruited in infancy receive two doses of PCV10, at 18 and 24 months of age. Control group participants recruited at 18 months of age receive a single dose of PCV10 at 24 months. Intervention group participants receive at least two doses of PCV in the first year of life. All participants receive pneumococcal immunisation that is likely to be effective and is not otherwise available in Vietnam. The specific regimens to be evaluated are based on likely future global

**Table 1** Schedule of enrolment, interventions and assessments

| Age (months) | 2 | 3 | 4 | 5 | 6 | 7 | 9 | 9.5 | 10 | 12 | 18 | 19 | 24 |
|---|---|---|---|---|---|---|---|---|---|---|---|---|---|
| **Enrolment** | | | | | | | | | | | | | |
| Informed consent | X | | | | | | | | | | X* | | |
| Eligibility assessment | X | | | | | | | | | | X* | | |
| Allocation | X | | | | | | | | | | | | |
| **Interventions** | | | | | | | | | | | | | |
| PCV10—group A | X | X | X | | | | X | | | | | | |
| PCV10—group B | X | X | X | | | | | | | | | | |
| PCV10—group C | X | | X | | | | | X | | | | | |
| PCV10—group D | X | | | | X | | | | | | | | |
| PCV13—group E | X | | X | | | | | X | | | | | |
| PCV10—group F | | | | | | | | | | | X | | X |
| PCV10—group G | | | | | | | | | | | | | X |
| **Assessments** | | | | | | | | | | | | | |
| Demographics | X | | | | | | | | | | X* | | |
| Household characteristics | X | | | | | | | | | | X* | | |
| Nasopharyngeal swab | X | | | | X | | X | | | X | X | | X |
| Blood sample—group A | X† | | | X | | | X | | X | | X† | | |
| Blood sample—group B | | | | X | X | | X† | | X | | X† | | |
| Blood sample—group C | | | | X | X† | | X | | X | | X† | | |
| Blood sample—group D | | X | | | X | X | X† | | | | X† | | |
| Blood sample—group E | | X† | | X | | | X | | X | | X† | | |
| Blood sample—group F | | | | | | | | | | | X | X | X |
| Blood sample—group G | | | | | | | | | | | X | X | X |
| General health | X | X | X | X | X | X | X | X | X | X | X | X | X |

*Group G only. Any events occurring before 18 months do not apply to group G.

†Each participant provides only one of these blood samples (the last 50 participants per group enrolled into groups A–E provide this blood sample at 18 months; the remainder provide it at the other time point).

recommendations and to directly compare the two licensed PCVs.

Both PCV10 and PCV13 have been shown to be non-inferior to PCV7 for the serotypes common to both vaccines, and to have the potential to provide protection against the additional serotypes included.[9–11] For both vaccines, the most common adverse reactions are redness at the injection site and irritability, which are common following administration of other vaccines. Other adverse reactions may include drowsiness; temporary loss of appetite; pain, redness or swelling at the injection site; and fever. Such reactions are usually temporary.

## OBJECTIVES
This trial has been designed to answer two independent questions concurrently, relating to the evaluation of different schedules incorporating PCV10 and the comparison of PCV10 and PCV13:
1. What is the optimal schedule for provision of EPI vaccines with the incorporation of PCV10; and
2. How do the responses to vaccination with PCV10 or PCV13 compare?

The primary endpoint for both study questions is the post-primary series immunogenicity. For this endpoint, data from arms A and B are combined, as they receive an identical three-dose primary series (see table 1 for a detailed description of the trial arms). The primary analysis for each study question is to assess non-inferiority of the post-primary series immunogenicity (in terms of the proportion of participants achieving protective levels of serotype-specific IgG of ≥0.35 µg/mL), using arms A+B as the comparator (see below for details). Non-inferiority is assessed for each of the 10 serotypes in PCV10, and an overall conclusion of non-inferiority drawn if found for at least 7 of the 10 serotypes.

### What is the optimal schedule for provision of Expanded Program of Immunisation (EPI) vaccines with the incorporation of PCV10?
#### Primary objective
The primary objective is to compare a 2+1 schedule at 2, 4 and 9.5 months of age with a 3+1 schedule at 2, 3, 4 and 9 months of age. The primary hypothesis is that the proportion of participants with protective levels of antibody is non-inferior following a two-dose primary series

(arm C) compared with a three-dose primary series (arms A+B). The schedules will also be compared in relation to the geometric mean concentrations (GMCs) of IgG and opsonophagocytosis post-primary series; the proportion of participants with protective levels of antibody, the GMCs of IgG and opsonophagocytosis post-booster; the memory B-cell responses; and the impact on nasopharyngeal (NP) carriage rates and density of bacteria of interest.

### Key secondary objectives

► To investigate an experimental two-dose schedule at 2 and 6 months of age (arm D), compared with a 3+1 schedule (arm A±B) and a 2+1 schedule (arm C); and

► To assess the impact of a booster dose on NP carriage of pneumococcus and NTHi, comparing a 3+1 schedule (arm A) with a 3+0 schedule (arm B) and with unvaccinated controls (arm F).

### How do the responses to vaccination with PCV10 or PCV13 compare?
#### Primary objective
The primary objective is to compare a PCV13 schedule at 2, 4 and 9.5 months of age with a PCV10 schedule at 2, 3, 4 and 9 months of age. The primary hypothesis is that the proportion of participants with protective levels of antibody is non-inferior following a two-dose primary series of PCV13 (arm E) compared with a three-dose primary series of PCV10 (arms A+B). The schedules will also be compared in relation to the GMCs of IgG and opsonophagocytosis post-primary series; the proportion of participants with protective levels of antibody, the GMCs of IgG and opsonophagocytosis post-booster; the memory B-cell responses; and the impact on nasopharyngeal (NP) carriage rates and density of bacteria of interest.

### Key secondary objectives

► To compare PCV10 (arm C) and PCV13 (arm E) in a 2+1 schedule at 2, 4 and 9.5 months of age; and

► To compare the responses to a single dose of PCV10 (arm D) and PCV13 (arm E).

### Additional objectives
Additional objectives relating to the second control group (arm G) are:

► To evaluate a single dose of PCV10 at 18 months of age, comparing serotype-specific antibody levels in arms F and G at 18, 19 and 24 months of age; and

► To compare the immunogenicity and reactogenicity of *Infanrix-hexa* at 18 months of age in children who have received three doses of *Infanrix-hexa* or *Quinvaxem* in infancy (arm G).

### Trial design
The Vietnam Pneumococcal Project is a single-blind, open-label, randomised controlled phase II/III non-inferiority trial to investigate simplified childhood vaccination schedules that are more appropriate for developing country use. This is a seven-arm trial that includes six different infant vaccination schedules (arms A–F) and an additional control group (arm G) recruited at 18 months of age (table 1). Arm A receives PCV10 at 2, 3, 4 and 9 months of age (3+1); arm B receives PCV10 at 2, 3 and 4 months (3+0); arm C receives PCV10 at 2, 4 and 9.5 months (2+1); arm D receives PCV10 at 2 and 6 months (two-dose); arm E receives PCV13 at 2, 4 and 9.5 months (2+1); arm F receives two doses of PCV10 at 18 and 24 months; and arm G receives one dose of PCV10 at 24 months. Participants also receive *Infanrix-hexa* (DTP-Hib-HBV-IPV) instead of the routine EPI vaccine *Quinvaxem* (DTP-Hib-HBV): four doses for participants in arms A–F and one dose for arm G participants.

## METHODS AND ANALYSIS
### Study setting
PCV introduction in Asia has been slow, in part due to a lack of local or regional data on the effect of PCV. We selected the Southeast Asian country of Vietnam as the location for the trial as a country with a strong health system, a track record of conducting relevant clinical trials, and a Government with strong interest both in the trial and in introducing PCV in the near future. Furthermore, trial results from Vietnam are likely to be considered as applicable to other countries in the region. This is the first trial involving infants to take place within Ho Chi Minh City, the largest city in Vietnam. The trial is conducted in two districts, District 4 and District 7. Districts are divided into communes, each of which has a health centre that provides preventive health services including EPI immunisations, along with some primary healthcare services. The study is conducted in one commune health centre in each district, with participants drawn from the surrounding communes within that district.

### Eligibility criteria
#### Inclusion criteria
Subjects must meet all of the following inclusion criteria in order to be eligible to participate: aged between 2 months and 2 months plus 2 weeks (arms A–F) or aged between 18 months and 18 months plus 4 weeks (arm G); no significant maternal or perinatal history; born at or after 36 weeks' gestation; written informed consent from the parent/legal guardian; lives within approximately 30 min of the commune health centre; anticipates living in the study area for the next 22 months (arms A–F) or 6 months (arm G); and received three doses of either *Quinvaxem* or *Infanrix-hexa* in infancy (arm G only).

#### Exclusion criteria
Subjects meeting any of the following exclusion criteria at baseline will be excluded from study participation: known allergy to any component of the vaccine; allergic or anaphylactic reaction to any previous vaccine; known immunodeficiency disorder; known HIV-infected mother; known thrombocytopenia or coagulation disorder; on immunosuppressive medication; administration or planned administration of any immunoglobulin or blood

product since birth; severe birth defect requiring ongoing medical care; chronic or progressive disease; seizure disorder; history of invasive pneumococcal, meningococcal or *H. influenzae* type b diseases, or tetanus, measles, pertussis or diphtheria infections; receipt of any 2-month vaccines through the EPI programme (arms A–F), or receipt of PCV (arm G); or family plans on giving the infant *Quinvaxem* (arms A–F).

## Interventions
### PCV schedules
Eligible participants recruited in infancy are randomised to one of six different vaccination schedules (table 1). Participants randomised to arms A–D receive PCV10 in a 3+1 schedule at 2, 3, 4 and 9 months of age; a 3+0 schedule at 2, 3 and 4 months of age; a 2+1 schedule at 2, 4 and 9.5 months of age; or a two-dose schedule at 2 and 6 months of age, respectively. Participants randomised to arm E receive PCV13 in a 2+1 schedule at 2, 4 and 9.5 months of age. Control group participants receive PCV10 at 18 and 24 months of age if randomised to arm F, or PCV10 at 24 months of age if recruited to arm G at 18 months of age. PCV is administered by intramuscular injection into the anterolateral thigh in children less than 18 months old and in the deltoid muscle of the arm in children aged 18 months and over. All vaccinations are performed by nurses specifically trained in infant vaccine administration.

### PCV10
PCV10 (*Synflorix*) is a 10-valent pneumococcal polysaccharide conjugate vaccine using protein D (a highly conserved surface protein from NTHi) as the main carrier protein. PCV10 is presented as a turbid white suspension in a two-dose phial. One dose consists of 0.5 mL of the liquid vaccine, containing 1 µg of pneumococcal polysaccharide from serotypes 1, 5, 6B, 7F, 9V, 14 and 23F and 3 µg of pneumococcal polysaccharide from serotypes 4, 18C and 19F. Serotypes 1, 4, 5, 6B, 7F, 9V, 14 and 23F are conjugated to protein D; serotype 18C is conjugated to tetanus toxoid carrier protein; and serotype 19F is conjugated to diphtheria toxoid carrier protein.

### PCV13
PCV13 (*Prevnar-13*) is a 13-valent pneumococcal polysaccharide conjugate vaccine using non-toxic diphtheria $CRM_{197}$ carrier protein. PCV13 is presented as a 0.5 mL suspension in a single-dose pre-filled syringe. One dose contains approximately 2.2 µg of pneumococcal polysaccharide from serotypes 1, 3, 4, 5, 6A, 7F, 9V, 14, 18C, 19A, 19F and 23F and 4.4 µg of pneumococcal polysaccharide from serotype 6B.

### Criteria for discontinuing or modifying allocated interventions
There is no modification of doses for participants in this study. If a participant has an allergic or anaphylactic response to vaccination, they will be withdrawn from the study. Participants may also be withdrawn voluntarily by the parent/legal guardian at any time, or by the study staff if they refuse any further study procedures or develop any of the exclusion criteria during the course of the study.

### Strategies to improve and monitor adherence
Scheduled visit dates are noted on a health record card kept by the parent. If a participant does not attend a scheduled visit, a reminder phone call is made from the study clinic. If the participant cannot be contacted directly, their local commune health centre is contacted for further follow-up by phone or by home visit.

### Relevant concomitant care
Participants receive *Infanrix-hexa*, which is only available on the private market, instead of the routine EPI vaccine *Quinvaxem.* Participants in arms A–F receive four doses in one of the following schedules: 2, 3, 4 and 19 months (arms A and B); 2, 4, 9.5 and 19 months (arms C and E); 2, 4, 6 and 19 months (arm D); or 2, 3, 4 and 18 months (arm F); and participants in arm G receive one dose at 18 months of age. The routine EPI measles and measles–rubella immunisations are also provided during the course of the study: measles at 9 months of age and measles–rubella at 18 (arms A–E) or 19 (arms F–G) months of age. Participants allocated to one of the 2+1 vaccination schedules (arms C and E) receive measles at 9 months of age and receive PCV and *Infanrix-hexa* 2 weeks later. For visits with two vaccinations, the vaccines are administered in different limbs. Other vaccinations are permitted in this study with a 2-week interval from study vaccines, with the exception of *Quinvaxem* in arms A–F. Other medications are also permitted, with the exception of immunosuppressive medication and medications listed as contraindicated to the study vaccines.

## Outcomes
### Primary outcome measure
The primary outcome measure is the concentration of serotype-specific IgG for the 10 serotypes common to both PCV10 and PCV13, assessed 4 weeks post-primary series and measured using a modified third-generation standardised ELISA.[30] Primary comparisons between arms are made in terms of the proportion of children with antibody concentration ≥0.35 µg/mL for individual serotypes. The cut-off of 0.35 µg/mL was determined as a result of a pooled analysis of data from efficacy trials[31] and is used as the basis for non-inferiority assessments for the approval of new PCVs.[32–34]

### Secondary immunogenicity outcome measures
► Serotype-specific IgG antibody concentrations for all PCV13 serotypes are measured by ELISA from all blood samples (table 1) and are summarised in terms of both the proportion of children with antibody concentration ≥0.35 µg/mL and the GMC.
► Opsonisation indices (OIs) for all PCV13 serotypes are measured by opsonophagocytic assay (OPA)[35] for 100 participants per intervention group (arms A–E) 4 weeks post-primary series and 4 weeks post-booster,

and are summarised in terms of the proportion of participants with OI ≥8 and the geometric mean titre.

▶ Polysaccharide-specific memory B cells for serotypes 1, 5, 6B, 14, 18C, 19A and 23F are enumerated by ELISPOT[35] for 50 participants per intervention group (arms A–E) post-booster and at 18 months of age, and for 100 participants per control group (arms F and G) at 18 and 24 months of age. The results are summarised as the median number of antibody-secreting cells.

### Nasopharyngeal carriage outcome measures

▶ NP carriage of pneumococcal serotypes is measured by traditional culture (colonial morphology, α-haemolysis, the optochin test and *lytA* PCR where indicated)[36] and latex agglutination using type-specific antisera at 2, 6, 9 and 12 months of age in all groups and at 18 and 24 months of age in the control groups (arms F and G). NP carriage and density of pneumococcal serotypes are measured by quantitative real-time PCR (qPCR) targeting *lytA* and microarray at 18 and 24 months of age.[37 38] Overall, capsular, vaccine-type and serotype-specific carriage rates are described. The antimicrobial resistance of pneumococcal isolates is determined at 12 months of age by the Clinical and Laboratory Standards Institute (CLSI) disk diffusion method, for oxacillin, erythromycin, trimethoprim/sulfamethoxazole, ofloxacin, clindamycin, vancomycin, tetracycline and chloramphenicol. E-tests are conducted for penicillin, ceftriaxone and vancomycin where indicated, and CLSI breakpoints applied.

▶ NP carriage of *H. influenzae* is measured by traditional culture (colonial morphology, X and V dependence, *SiaT* PCR for discrimination from *H. haemolyticus* and the Phadebact Haemophilus coagglutination test) at 12 months of age in all groups, at 6 and 9 months of age in arms A and C, and from all swabs in the control groups (arms F and G). Overall density of *H. influenzae* carriage is measured by qPCR targeting *hpd* and *SiaT* diagnostic targets at 18 and 24 months of age.[39 40]

### Immunogenicity of *Infanrix-hexa*

Immunogenicity of *Infanrix-hexa* is measured in terms of IgG levels to diphtheria, tetanus, Hib PRP antigen, hepatitis B surface antigen and *Bordetella pertussis*. IgG levels will be determined by ELISA, using commercial test kits.

An overview of the procedures for collection, transportation and laboratory analyses of the blood and NP samples can be found in online supplementary appendix 1.

### Sample size

The target sample size for infant recruitment (groups A–F) is 1200 with an allocation ratio of 3:3:5:4:5:4, resulting in target group sizes of A=150, B=150, C=250, D=200, E=250 and F=200. An additional target of 200 children aged 18 months are recruited into group G. Sample size

calculations are based on the primary outcome of post-primary series immunogenicity (proportion of participants with serotype-specific antibody concentrations ≥0.35 µg/mL) for each of the two study questions. A non-inferiority margin of a 10% difference in absolute risk is deemed clinically significant, as used by regulatory authorities. Non-inferiority is assessed for each of the 10 serotypes in PCV10 (comparing groups A+B with group C or group E), and an overall conclusion of non-inferiority is drawn if the alternative hypotheses are accepted for at least 7 of the 10 serotypes. This sample size provides >99% power for the overall conclusion of non-inferiority with a 5% one-sided type I error rate, estimated by simulation using a tailor-made program written for implementation in Stata with 10 000 replications.[41] Powers for serotype-specific hypotheses range from 83% to >99%, calculated in PASS Software 2002 using the Farrington-Manning (1990) method.[42] Based on findings from our earlier work in Fiji and from data available in the literature,[43–45] the assumed probabilities of antibody concentration ≥0.35 µg/mL are 95% for serotypes 1, 4, 5, 7F, 9V, 14 and 19F; 90% for serotype 18C; 80% for serotype 23F; and 75% for serotype 6B. The within-subject correlation between the multiple binary endpoints is captured by a subject-level variation term with SD 1.7 in a random-effect logistic regression model, and the loss to follow-up rate is assumed to be 5% post-primary series and 10% at 12 months of age. The sample size also provides 98% power to detect a difference in post-primary series immunogenicity following two doses of PCV10 or PCV13, defined by a 10% difference in absolute risk based on Fisher's exact test (5% two-sided).

### Carriage outcomes

The sample size provides 76% and 71% power to detect a difference in NTHi carriage rates at 12 months of age between groups A and F and groups A and B, respectively, and 64% and 59% power to detect a difference in vaccine-type pneumococcal carriage rates between groups A and F and groups A and B, respectively. Difference in carriage is defined by a relative risk of 0.6. The calculations were based on Fisher's exact tests (5% one-sided), assuming carriage rates in group F (controls) of 30% for NTHi and 24% for vaccine-type pneumococci, based on data from Vietnam (L Yoshida, personal communication).

### Recruitment

Participants in groups A–F are recruited from infants born in the study communes during the enrolment period. Commune health centre staff identify potential participants from the commune health centre birth records. Based on the expected number of births, around a quarter of infants born in the study communes need to be enrolled to complete recruitment within the target enrolment period of 12 months. Recruitment rates will be monitored on a monthly basis and meetings held with study staff and commune health centre staff to discuss any significant declines in recruitment rates. Commune health centre staff visit the home of potential participants

when the infant is approximately 6 weeks old and provide verbal and written information about the trial, in Vietnamese. Those interested in participating are referred to the study clinic when the infant is approximately 2 months old. At this time, written informed consent is obtained (online supplementary appendix 2), after which a study nurse/doctor examines the infant to ensure that all the eligibility criteria are met. Participants in group G are recruited from children turning 18 months old in the study communes in parallel to the children in groups A–F turning 18 months.

### Allocation

The allocation sequence for groups A–F is produced using a computer-generated list of random numbers using a block randomisation scheme, stratified by district. The group allocation is contained within a sealed envelope at the study clinic, with sequential ID numbers written on the outside of the envelope. The allocation sequence is generated at Menzies School of Health Research. A study doctor will enrol participants and assign them to a study group by selecting the next available envelope. The envelope is not opened until after completion of the informed consent and eligibility assessment processes.

### Blinding

All laboratory staff are blinded to the study group allocation as the key outcome measures that address the study objectives are all laboratory based. Laboratory samples are labelled with the ID number, which does not identify the study group. Given the different timing of the vaccination schedules in the different groups, the study nurses, vaccine administrators and participants will not be blinded to the study group allocation.

### Data collection methods

Standardised carbon copy data collection forms are used and are completed by dedicated, trained study staff. The original is transported to the trial office for data entry, with the carbon copy filed at the clinic. Blood samples and NP swabs are collected by staff specifically trained in the collection of samples from infants, and the volume of blood collected and the swab quality are recorded.

Retention: Appointments are documented on a parent-held health record card and a reminder phone call made the week before the scheduled visit. If a participant fails to attend an appointment, a follow-up phone call is made to rebook the visit. Participants are given a small payment towards the transport costs of coming to the clinic for each study visit. Participants who miss a study visit will continue to be followed up for both sample collection and vaccine administration where possible, with attempts made to contact them until such time as they would have completed the study.

### Data management

Data collection forms are double-entered by dedicated data entry staff into pre-coded EpiData V.3.1 files with built-in range and consistency checks. Entered data are validated monthly and then uploaded to a central Microsoft Access database, stored on a secure server. Immunology results are double-entered in a Microsoft Excel spreadsheet. NP culture results are entered in a Microsoft Access database and qPCR and microarray results exported from SentiNET into a Microsoft Excel database. The data collection forms and laboratory results are linked at the time of analysis.

### Statistical methods
#### Analysis of primary and secondary outcomes

For each of the two study questions, the primary objective is to compare a 2+1 schedule of (1) PCV10 and (2) PCV13, with a 3+1 schedule of PCV10. The primary outcome is the proportion of participants with serotype-specific antibody concentrations ≥0.35 µg/mL, 4 weeks post-primary series (at 5 months of age). Data from arms A and B are combined to form the three-dose post-primary series group. The primary analyses assess the non-inferiority of (1) two doses of PCV10 at 2 and 4 months of age (arm C) compared with three doses at 2, 3 and 4 months of age (arms A+B); and (2) two doses of PCV13 at 2 and 4 months of age (arm E) compared with three doses of PCV10 at 2, 3 and 4 months of age (arms A+B). The proportion of children achieving protective levels of serotype-specific IgG (≥0.35 µg/mL) 4 weeks post-primary series is determined for each of the 10 PCV10 serotypes. The non-inferiority margin is defined by a 10% difference in absolute risk. The serotype-specific risk differences (arms A+B/ arm C) with 90% CIs are calculated using the Newcombe Score method and the null hypothesis rejected if the upper bound of the CI is <10%. Overall non-inferiority is declared if at least 7 of the 10 individual null hypotheses are rejected at a one-sided 5% level of significance. Secondary data analyses to address the primary objective include the ratio of GMCs post-primary series (arm C/arms A+B and arm E/arms A+B) with 95% CIs, and the booster response analysed by analysis of covariance, adjusting for pre-booster levels.

#### Analysis of key secondary objectives for study question 1

► A single dose of PCV10 at 2 months of age (arm D) will be assessed for non-inferiority to three doses at 2, 3 and 4 months of age (arms A+B), as described for the primary objective.
► The impact of a booster dose on pneumococcal and NTHi carriage will be assessed at 12 months of age. Overall pneumococcal, capsular pneumococcal, PCV10 type and NTHi carriage rates will be determined. Proportions will first be compared between the 3+1 group (arm A) and the control group (arm F), using Fisher's exact test. Where significant differences are found, rates will then be compared between the 3+0 group (arm B) and controls and between the 3+1 and 3+0 groups.

#### Analysis of key secondary objectives for study question 2

► The immunogenicity of two doses of PCV10 or PCV13 will be compared in relation to the proportion of

participants with serotype-specific antibody concentrations ≥0.35 µg/mL (to the 10 shared serotypes), 4 weeks post-primary series (at 5 months of age). A significant difference will be indicated by a 10% difference in absolute risk, comparing PCV10 (arm C) with PCV13 (arm E), and an overall difference will be declared if at least 7 of the 10 individual null hypotheses are rejected and the seven differences are in the same direction.

► The immunogenicity of a single dose of PCV10 or PCV13 will be compared, as described for the immunogenicity of two doses.

### Additional analyses
Descriptive analyses at the group level will be conducted on the OPA, ELISPOT and microarray data.

### Populations of analysis
Analyses will be on a per-protocol population. The primary non-inferiority analyses will be repeated on an intention-to-treat population (ITT), with all participants analysed in the group they were randomised to. Any differences between the per-protocol and ITT analyses will be reported. For each outcome, all available data will contribute to the analyses. To investigate whether data are missing completely at random, we will explore whether attrition varies across the study arms based on baseline covariates. If differential attrition is dependent on baseline variables, we will use a modelling approach to adjust for any such baseline factors and we will present the adjusted results along with the primary analysis.

### Additional populations of analysis
► OPAs will be conducted on a subset of 100 participants per group. The first 100 participants per group with both post-primary series and post-booster blood samples available will contribute to the OPA analysis.

► B-cell assays will be conducted on a subset of 50 participants per group for arms A–E and 100 participants per group for arms F and G. The last 50/100 participants enrolled per group will have blood samples collected for the B-cell analysis.

### Data monitoring
Data monitoring committee: Safety oversight is under the direction of an independent Data Safety and Monitoring Board (DSMB), in accordance with a DSMB Charter kept in the trial office. The DSMB will meet approximately three times a year to review aggregate and individual participant data related to safety, data integrity and overall conduct of the trial, including a detailed review of all serious adverse events (SAEs).

Interim analyses and stopping guidelines: No interim analyses are planned. Stopping guidelines are based on safety. An extraordinary meeting of the DSMB will be called in the event that serious safety issues emerge, to provide recommendations regarding termination of the trial. A final decision to terminate rests with the Principal Investigators and the Sponsor.

### Harms
Data on SAEs will be collected throughout the study, with parents asked about hospitalisations and significant signs and symptoms at each study visit and through a regular review of hospital records. Details of any SAEs will be recorded on the standard reporting form from the Vietnam Ministry of Health and reported to the Principal Investigators and the Ethics Committees. Participants will be kept under observation for 30 min following vaccine administration to monitor for any adverse reactions, and information on reactogenicity in the 72 hours following vaccine administration will be recorded on parent held diary cards.

### Auditing
External site monitoring will be provided by FHI360, to independently assess protocol and good clinical practice (GCP) compliance. Monitoring visits will occur at study initiation, close-out and approximately twice a year in each study clinic. 100% of Informed Consent Forms and SAEs and a random selection of approximately 20% of participant folders will be monitored, along with the Trial Regulatory File and laboratory records.

### Patient and public involvement
Patients were not involved in the development, design, recruitment or conduct of the study. Community consultation took place at the district level during the design phase, as well as discussion and approval of the design from the district and city level Ministry of Health and the People's Committee of Ho Chi Minh City. Participants will be informed of the overall study results by post, with a postal address collected at the final study visit.

## ETHICS AND DISSEMINATION
### Research ethics approval
The protocol, the Plain Language Statement (PLS) and the Informed Consent Form (ICF) have approval from the Institutional Review Board at the Pasteur Institute of Ho Chi Minh City, the Vietnam Ministry of Health Ethical Review Committee and the Human Research Ethics Committee of the Northern Territory Department of Health and the Menzies School of Health Research. Both Ethics Committees receive annual reports on the trial progress, for continuing approval of the trial.

### Protocol amendments
Any modifications to the protocol that may impact on the conduct of the study will be documented in a formal protocol amendment and approved by both Ethics Committees prior to implementation of the changes. The modified protocol will be given a new version number and date. The Ethics Committees will also be notified of any minor corrections/clarifications or administrative changes to the protocol, which will be documented in a protocol amendment letter. Significant protocol changes will also be updated in the ClinicalTrials.gov record.

## Consent

### Obtaining consent

The consent process is undertaken by specifically trained study staff. The study staff will go through the PLS and ICF, translated into Vietnamese, in detail with the potential participant's parent/legal guardian. The study staff will then discuss the trial further and answer any questions that may arise. Written informed consent is required prior to enrolment of the infant into the study. Consent is obtained from the parent/legal guardian as the participants are too young to provide consent themselves. A copy of the PLS and ICF will be given to the parent/legal guardian for their records.

### Ancillary studies

Specific consent for the indefinite storage of blood and NP samples for future research related to the trial will be obtained from the parent/legal guardian and recorded on the ICF. Any future research will undergo ethical review. Any samples for which indefinite storage is not consented to will be destroyed at the close of the trial.

## Confidentiality

All study-related information will be stored securely and held in strict confidence. All documents kept at the study clinics, including the ICFs and participant folders, are stored in locked cabinets. All documents kept centrally are stored in the trial office, which is kept locked. Electronic data is stored in the trial office and on a secure password protected server. The electronic data and laboratory samples are coded by a unique participant number and do not contain the participant name. Access to participants' information will be granted to FHI360 for monitoring purposes, and to the Ethics Committees or DSMB if required.

## Access to data

The final trial dataset will be under the custody of the trial sponsor, Murdoch Children's Research Institute (MCRI). The Principal Investigator, trial manager and trial statistician will have access to the full anonymised final dataset.

## Ancillary and post-trial care

Participants are advised to come to the study clinic for ancillary care, or to Children's Hospital Number 2 in Ho Chi Minh City, where they will not be charged for treatment and services. All participants are covered by clinical trials insurance for trial related harms.

## Dissemination policy

### Plans

Participants will be informed of the overall study results by post, with a postal address collected at the final study visit. Following completion of the trial, the results will be submitted for publication in peer-reviewed journals, and presented at relevant international conferences.

Agreements between MCRI and each of the Pasteur Institute of Ho Chi Minh City and GSK Biologicals SA provide that a party must obtain the prior approval of the other parties in advance of submitting a manuscript for publication, and that such approval will not be unreasonably withheld.

### Authorship

A publication subcommittee will consider all proposed publications, with the final decision on content and authorship resting with the Principal Investigator. The role of each author will be published. Group authors may be used where appropriate. There are no plans for the use of professional writers.

### Reproducible research

There are no plans to grant public access to the full protocol, participant-level dataset or statistical code.

**Author affiliations**

[1]Global Health Division, Menzies School of Health Research, Darwin, Northern Territory, Australia

[2]Epidemiology and Population Health, London School of Hygiene and Tropical Medicine, London, UK

[3]Pneumococcal Research, Murdoch Children's Research Institute, Melbourne, Victoria, Australia

[4]Department of Disease Control and Prevention, Pasteur Institute of Ho Chi Minh City, Ho Chi Minh City, Viet Nam

[5]Centre for Quantitative Medicine, Duke-NUS Medical School, Singapore

[6]Centre for Child Health Research, University of Tampere and Tampere University Hospital, Tampere, Finland

[7]Department of Paediatrics, University of Melbourne, Melbourne, Victoria, Australia

[8]Child Health Division, Menzies School of Health Research, Darwin, Northern Territory, Australia

[9]Microbiology and Immunology, Pasteur Institute of Ho Chi Minh City, Ho Chi Minh City, Viet Nam

**Acknowledgements** The authors thank the study participants and their families.

**Contributors** BT was involved with the study design, led the funding and ethics applications, has been involved in the day-to-day management of the trial and data analysis, and drafted the protocol and this manuscript. NTT advised on the study design and location, was involved in the approval processes in Vietnam and has been involved in the day-to-day management and implementation of the trial. DYU advised on the study design and location and has been involved in the day-to-day implementation of the trial. AB advised on the study design, assisted with the funding applications, and advised on and provided oversight of the immunology laboratory procedures. KB advised on the study design and location and has been responsible for the day-to-day management and implementation of the trial. YBC advised on the study design and funding applications, especially the statistical aspects of the trial. PL advised on the study design, assisted with the funding applications, and advised on and provided oversight of the immunology laboratory procedures. CDN advised on the study design and statistical analysis plan. NTMP advised on the study design and location, was involved in the approval processes in Vietnam and has been involved in the day-to-day management of the trial. CS advised on the study design, assisted with the funding applications, and advised on and provided oversight of the microbiology laboratory procedures. HS-V advised on the study design, assisted with the funding applications, and advised on and provided oversight of the microbiology laboratory procedures. TQHV advised on the study design and advised on and provided oversight of the laboratory procedures at Pasteur. TNH advised on the study design and location, undertook consultations, was involved in the approval processes in Vietnam and has had overall responsibility for the conduct of the trial in Vietnam as Site Principal Investigator. EKM conceived the study, undertook consultations, provided oversight for the funding and ethics applications, provided oversight for the conduct of the trial and data analysis, and has had overall responsibility for all aspects of the trial as the Principal Investigator. All authors contributed to refinement of the study protocol and reviewed and approved this manuscript.

**Funding** This work is supported by the National Health and Medical Research Council of Australia (grant no. 566792) and the Bill and Melinda Gates Foundation (grant no. OPP1116833). The doses of PCV10 and funding for the opsonophagocytic assays are provided by GlaxoSmithKline Biologicals SA (GSK).

**Competing interests** All authors receive salary support from grants from the National Health and Medical Research Council of Australia and/or the Bill and Melinda Gates Foundation. Non-financial support (in the form of PCV10 vaccine doses) and funding for opsonophagocytic assays are provided by GSK Biologicals SA. EKM is a member of the DSMB for a current Novavax trial, for which he receives consulting fees. He has received travel costs from the GSK group of companies for one international conference and an honorarium from Merck for one advisory group meeting. He does not have any paid consultancies with or receive any research funds from pharmaceutical companies. Members of CS's team have received awards that were funded (but not assessed) by Pfizer. None of the authors have any other competing interests to declare.

**Patient consent** Guardian consent obtained.

**Ethics approval** Vietnam Ministry of Health Ethics Committee and the Human Research Ethics Committee of the Northern Territory Department of Health and Menzies School of Health Research.

**Provenance and peer review** Not commissioned; externally peer reviewed.

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
