## [Reviewer comments · BMJ Open]

ARTICLE DETAILS

TITLE (PROVISIONAL)	Evaluation of different infant vaccination schedules incorporating pneumococcal vaccination (the Vietnam Pneumococcal Project): protocol of a randomised controlled trial
AUTHORS	Temple, Beth; Toan, Nguyen; Uyen, Doan; Balloch, Anne; Bright, Kathryn; Cheung, Yin Bun; Licciardi, Paul; Nguyen, Cattram; Phuong, Nguyen; Satzke, Catherine; Smith-Vaughan, Heidi; Vu, Thi Que Huong; Huu, Tran; Muholland, Kim

VERSION 1 – REVIEW

REVIEWER	Mtthew Snape University of Oxford, UK.
REVIEW RETURNED	19-Nov-2017

GENERAL COMMENTS	Thank you for the opportunity to review this protocol which details the Vietnam Pneumococcal Project, an ambitious and important randomised, single-blind controlled trial aiming to determine the optimal schedule of pneumococcal vaccination in low income settings with a high prevalence of pneumococcal disease. One of the major obstacles to introduction of pneumococcal vaccine into low and middle income settings is the prohibitive price, exacerbated by the fact that vaccine recipients require multiple doses of vaccine. Determining the lowest possible number of doses of vaccine needed to achieve protection from pneumococcal disease could mean that many more countries may be able to introduce it into their routine vaccination schedule, with the positive follow on effects for the general population. This study has been deigned to answer the following questions: what is the optimal schedule for provision of routine vaccines with the incorporation of PCV10; and how do the responses to vaccination with PCV10 or PCV13 compare? The study protocol specifically examines differences in post primary series immunogenicity (around five months of age) achieved by several different schedules of PCV10 and one of PCV13, compared with two control groups. The study looks extensively at different schedules of PCV10, but only one of PCV13, and the reasons for this are not fully addressed. The authors will be aware that the UK JCVI has recently announced that it will be recommending a switch to a 3, 12 month schedule for PCV13 based on the results of the Sched 3 study, demonstrating similar or higher IgG concentrations in the 1+1 group compared to 2+1 for 9 out of 13 serotypes. These results will be published in the
---

	very near future and it would be appropriate to reference these in this publication. It is notable that study participants were only recruited from areas within Ho Chi Minh City, and not in any regional areas. If the study authors are looking at the roll out of vaccines in low and middle income countries, then this is a potential weakness of the study design that could be discussed. The authors have said that one of the main limitations of the study is the fact that the nasopharyngeal data could be subject to natural variations in carriage, which could skew the data. However, all studies on nasopharyngeal carriage would be subject to this issue, and this is therefore not a limitation of this study specifically. The description of the study groups and objectives in the abstract is quite awkward; these are complicated but some rewording here would be appropriate. There could also be a bit more precision in the language, e.g. line 31 page 7: 'Both have been shown to be non-inferior to PCV-7'....on what measures? Likewise describing the second primary objective as 'How do the responses to vaccination with PCV10 or PCV13 compare' is too vague. Is this referring to immune responses, (presumably serotype specific IgG as measured by ELISA), or reactogenicity? The endpoints on page 11 make clear that this is referring to the immune response...but it is unusual to have so many endpoints – GMCs, % > 0.35, OPA and memory B cells. Which is the primary endpoint?
--	--

REVIEWER	Maria Deloria Knoll International Vaccine Access Center (IVAC) at the Johns Hopkins Bloomberg School of Public Health, USA
REVIEW RETURNED	27-Dec-2017

GENERAL COMMENTS	This is a protocol submission describing a randomized trial to evaluate the immune response to various dosing schedules of two PCV products administered to Vietnamese infants and young children. The investigators have described the trial following the Standard Protocol Items: Recommendations for Interventional Trials (SPIRIT) 2013 recommendations. The investigators have provided sufficient detail in this protocol description to give readers a clear picture of what the trial will entail and the nature of the results that will likely be forthcoming. They very closely followed the format and addressed the content of SPIRIT recommendations. Below are the few items that were included in the SPIRIT recommendations that were not addressed by the authors, or that needed more detail, as well as some suggestions/observations regarding the proposed methods.  1. Please consider changing the control group label “No intervention F”, to “Experimental F” since a non-standard dose of PCV at age 18m is being evaluated in these children in one of the secondary analyses in comparison to a group of children who have not (yet) been vaccinated (Control group G).
--

	2. OPA analyses: the proposed analyses (% indices ≥ 8 and GMTs) are standard OPA analyses and will facilitate comparison to results of other trials, but may not be biologically meaningful for some serotypes or subject to bias from extreme values due to wide heterogeneity of response among individuals. The ≥ 8 threshold is too low for several serotypes to be associated with protection. In conversations with David Goldblatt, it has been suggested to me that additional analyses that describe the distribution may be more informative to answer the research questions, such as comparing the highest value that 80% of the subjects have achieved. 3. Background: a. the protocol states “there have been no published studies to date directly comparing [PCV10 and PCV13] post-primary series immunogenicity or impact on nasopharyngeal (NP) carriage.” However, there have been three head-to-head trials of 3+0 or 2+1 schedules evaluating immunogenicity post-primary and two head-to-head NP carriage trials:  i. Temple, B., et al., HEAD-TO-HEAD COMPARISON OF PCV10 AND PCV13: POST-PRIMARY SERIES IMMUNOGENICITY AND IMPACT ON NASOPHARYNGEAL CARRIAGE AT 12 MONTHS OF AGE. ISPPD-10, 2016; ii. Pomat, W., et al., IMMUNOGENICITY OF 10-VALENT AND 13-VALENT PNEUMOCOCCAL CONJUGATE VACCINES GIVEN AT 1-2-3 MONTHS OF AGE IN PAPUA NEW GUINEAN INFANTS: A RANDOMISED CONTROLLED TRIAL. ISPPD-10, 2016; iii. Van, et al., Differential B-Cell Memory Around the 11-Month Booster in Children Vaccinated with a 10- or 13-Valent Pneumococcal Conjugate Vaccine. Clinical Infectious Diseases, 2015. 61(3): p. 342-349; iv. Wijmenga-Monsuur, A.J., et al., Direct Comparison of Immunogenicity Induced by 10- or 13-Valent Pneumococcal Conjugate Vaccine around the 11-Month Booster in Dutch Infants. PLoS One, 2015. 10(12): p. e0144739; v. Phan, T.V., et al., IMMUNOGENICITY AND MEMORY B CELL RESPONSE FOLLOWING ALTERNATIVE PNEUMOCOCCAL VACCINATION STRATEGIES IN VIETNAM. ISPPD-10, 2016). vi. Orami, T., et al., IMPACT OF 10 VALENT AND 13 VALENT PNEUMOCOCCAL CONJUGATE VACCINES ON PNEUMOCOCCAL CARRIAGE AMONG
--	--

PAPUA NEW GUINEAN INFANTS. ISPPD-10, 2016.

In addition, when combining head to head with single arm studies, for immunogenicity there were 63 PCV10 studies and 56 PCV13 studies and there were 9 single-arm NP carriage trials (8 PCV10, 1 PCV13) plus 18 observational arms of routine use (5 PCV10, 13 PCV13) for which meta-analyses were performed and reported comparing products: http://www.who.int/immunization/sage/meetings/2017/october/3_FULL_PRIME_REPORT_2017Sep26.pdf.

- b. The same recent systematic review compared 2-dose vs 3-dose primary schedules and while it was slightly more comprehensive (10 head to head RCTs) than what is described, the results were similar wrt antibody concentrations above the correlate of protection (i.e., little difference between schedules in the proportion of subjects with antibody concentrations above the correlate of protection except serotypes 6A and 6B). But it should be mentioned that the 2-dose primary schedule elicits lower post-primary GMC than 3-dose primary schedule for most vaccine serotypes and post-dose 3 GMCs are higher for infants receiving a 2+1 schedule than those receiving a 3+0 schedule for most serotypes.

4. Item 11:

- a. There is some lack of clarity re: whether or not the co-administered vaccines (measles and Infanrix-hexa) are being administered as per the routine EPI schedule for these children or whether one or more doses is being added. The protocol states that “the Vietnam Ministry of Health (MOH) does not permit the co-administration of measles vaccine and Infanrix-hexa vaccine, which was scheduled at 9 months of age in Arms C and E. An additional visit at 9.5 months of age was added for these groups, for receipt of PCV and Infanrix-hexa.” “Participants allocated to one of the 2+1 vaccination schedules (Arms C and E) receive measles at 9 months of age and receive PCV and Infanrix-hexa two weeks later.” This implies these children do not routinely receive these two vaccines at 9 months and therefore this study is deviating from routine care in that regard. If so, this should be made more clear in the proposal what is different for the children enrolled in this study vs. for children who do not enroll. If there is a difference, this should also be made clear in the consent form.
- b. There is also some lack of clarity re: which concomitant vaccines will be given at 9 months to Group A (3+1 PCV) and how and why this differs from Groups C and E (2+1 PCV). In Groups C and

	E the 9-month Infanrix-hexa immunization is being moved to 9.5m and given with PCV at 9.5m. So if PCV is being given to group A at 9 m and not 9.5m, then does this mean that Infanrix-hexa alone is being given at 9.5m since as mentioned above “the Vietnam MOH does not permit the co-administration of measles vaccine and Infanrix-hexa vaccine”? Or perhaps this is an error and PCV (and Infanrix-hexa) should also be a 9.5m for Group A? c. Specify the timing (ages) of the 4 Infanrix-hexa doses and specify whether this schedule is the same or different from the national EPI schedule being administered to children from the study area. 5. Item 12 (Secondary immunogenicity outcome measures): Are both pre-PCV blood draws at 18m and 19m necessary in children in Group G (who get 1 dose of PCV at 24m)? It seems only one ‘18m’ pre-PCV blood is needed, unless inter-subject heterogeneity needs to be assessed in unimmunized children, but that was not described in the proposal. Please justify the collection of these two pre-PCV blood draws. 6. Item 13 (timeline): a. please add the concomitant vaccines being administered as part of the study to the timeline (the 4 Infanrix-hexa doses, measles and measles-rubella) b. There are two blood draws in each group for which “each participant provides only one of these blood samples”. Please describe the assignment process wrt when and how assigned, e.g., is this part of the randomization assignment at the time of enrollment? 7. Item 14 (sample size): a. Only tests with 1-sided type I error are described. Please justify why the product comparison is not a test with a 2-sided type I error. b. Why is carriage assessed as a difference while immunogenicity is assessed as non-inferiority? c. The difference in carriage between groups appears to have the same relative risk (0.6), regardless of which two groups are being compared. I think this likely overestimates the effect size for the 3+1 vs 3+0 schedule comparison (Groups A vs B), and therefore overstates the power to detect a difference for this comparison. Please justify this assumption. 8. Item 15 (recruitment): this section is supposed to describe “Strategies for achieving adequate participant enrolment to reach target sample size”. Please describe elements that convey confidence in the ability of the trial to meet the sample size, such as expected recruitment rates based on previous similar studies, duration of the recruitment period and whether this could be extended if necessary, plans to monitor recruitment during the trial, and any financial or non-financial incentives provided to participants for
--	--

	enrolment/retention. Details for this particular trial that would be informative include describing how the link with parents will be maintained between the time of first contact at the child's birth until 2 months of age when they are approached for consent to ensure they do not previously receive EPI vaccines that would make them ineligible, and how and where they will be approached at 2m of age for enrollment procedures. 9. Item 18: Are the CRFs available for public viewing and if so where? 10. Item 20c (Definition of analysis population relating to protocol non-adherence): the protocol states that the [primary] analysis will be intent-to-treat, which biases findings towards similarity (i.e., non-inferiority) if there is incomplete immunization in the group assigned to more doses. And since the primary analysis is a non-inferiority analysis, this seems like a decision that has the potential to bias the results towards that finding. A more conservative approach seems to me to be a per-protocol analysis where the biological performance of 2 doses vs 3 doses could be compared, rather than the 'programmatic' evaluation of 2 doses being similar to 3 doses because kids fail to show for the 3rd dose. Please justify the choice for an ITT analysis over PP for the non-inferiority analyses and for all of the biological (lab-based) endpoints even if evaluating a difference instead of non-inferiority, given that the failure to show for a vaccination is unlikely to be associated with the immune response (i.e., the PP analysis is not likely to be biased). 11. Appendix 1 (Control information sheet): It says "Three blood tests will be taken... The blood tests are to check the response to the vaccines". However, in this group PCV is given at the time of the last blood draw and therefore will not be used to evaluate the response to PCV and there is no mention of testing responses to Infanrix-hexa, Measles or Rubella. Please justify the inclusion of this sentence in this consent form. Other minor items: 12. Abstract:  a. suggest changing "one of six PCV schedules" to "one of six PCV infant schedules" since response to PCV at 18m in Control Group F is being evaluated by comparing to no vaccine (Group G). b. suggest changing "unvaccinated controls that receive PCV10 and 18 and 24 months" (since that sounds contradictory) to "controls that receive PCV10 and 18 and 24 months" c. add as secondary outcome measures, response to serotypes that are in PCV13 but not PCV10. 13. Background:  a. "Three PCV schedules are currently in routine use around the world" – since a change from 2+1 to 1+1 is now being implemented in the UK (i.e., a 4th schedule), please consider a suggestion to clarify
--	---

	this sentence to reflect that there are 3 schedules used for introduction of PCV to distinguish from maintenance schedules that have fewer doses. b. The WHO currently recommends that the booster dose be given between the age of 9 and 15 months (http://www.who.int/ith/vaccines/pneumococcal/en/). Therefore, providing additional clarity would be useful regarding the statement “In developing countries, a 2+1 schedule with an earlier booster dose may be advantageous. This modified schedule ... could enable the booster dose to coincide with measles vaccination.” Perhaps you mean “... with the earliest recommended booster dose (i.e., at 9 months rather than at the commonly administered 12 or 15 months)”. c. There is a more up-to-date and more relevant (i.e., evaluating PCV13) publication regarding the data of a single dose following a booster compared to 2+1 to add to the current text (“a booster dose of the 23-valent pneumococcal polysaccharide vaccine at 12 months of age was more immunogenic following a single dose primary series of PCV7 compared with a two or three dose primary series”), but the findings were similar: Pneumococcal conjugate vaccine 13 delivered as one primary and one booster dose (1 + 1) compared with two primary doses and a booster (2 + 1) in UK infants: a multicentre, parallel group randomised controlled trial. Goldblatt D, Southern J, Andrews NJ, Burbidge P, Partington J, Roalfe L, Valente Pinto M, Thalasselis V, Plested E, Richardson H, Snape MD, Miller E. Lancet Infect Dis. 2017 Nov 22. pii: S1473-3099(17)30654-0. doi: 10.1016/S1473-3099(17)30654-0. d. The background states “Little is known about the effect of different PCV schedules on carriage” and cites two studies evaluating impact of 2- vs 3-dose schedules on VT carriage; however, a recent systematic review of 2+1 and 3+0 studies reports findings from 2 head-to-head trials (both PCV10), 4 trials evaluating 3+0 schedules, 3 trials evaluating 2+1 schedules and 18 observational arms (10 of 3+0 and 8 of 2+1) evaluating PCV10 or PCV13 impact on NP carriage in routine use. This review found no compelling evidence that one schedule consistently performed better over the other. http://www.who.int/immunization/sage/meetings/2017/october/3_FULL_PRIME_REPORT_2017Sep26.pdf. 14. Item 2b:
--	--

	 a. provide phone number and mailing address of the contact person b. provide title of the contact of scientific queries c. Suggest changing the health condition studied from “pneumococcal vaccination” to “immune response to pneumococcal vaccination” d. item #21 (ethics review process information) – please respond – I believe this is pending at this stage? e. Although items #22-24 pertain to results (Date of study completion, Date of posting of results, Plan to share participant-level data), which is not applicable to this study at this stage, planned or anticipated responses would be helpful and informative in the event this document is not updated. 15. #7:  a. page 10, primary obj 1 statement: It is more clear to specify ‘geometric mean IgG’ instead of just ‘IgG levels’ to distinguish from proportion with protective levels of antibody. b. page 11, primary obj 2 statement: regarding “the primary hypothesis that the immunogenicity is non-inferior”, it is more clear to add “as measured by the proportion with protective levels of antibody”. 16. Item 11: specify whether PCV will be given in a different limb from concomitant vaccines. 17. Item 20a (stats): the protocol states that the booster response will be analyzed by ANCOVA – what are the planned covariates in this model that will be adjusted for? 18. Item 20 (bottom of page 22): the protocol states that “Further details of the planned statistical methods can be found in the Statistical Analysis Plan.” Please indicate where this can be found. 19. Item 25: please indicate whether or not there are plans to communicate protocol updates to the trial registry/journal. 20. Item 29: access to the data: the protocol states that the ‘trial manager’ will have access to the full anonymized final dataset. Who is the trial manager and where are they located (i.e., what organization)? This statement implies no one else has access, including none of the study PIs, data analysts, statisticians – can you clarify? 21. Appendix 2: please describe where and how specimens be kept for long-term storage for future potential analyses/studies.
--	--

REVIEWER	Gerhard Falkenhorst, PhD Robert Koch Institute, Berlin, Germany Department of Infectious Disease Epidemiology
REVIEW RETURNED	03-Jan-2018

GENERAL COMMENTS	This manuscript is a detailed description of the study protocol of an ongoing randomized trial in Vietnam comparing different infant vaccination schedules using either PCV10 or PCV13. The trial evaluates several immunogenicity endpoints (serotype-specific IgG and opsonisation indices, and number of polysaccharide-specific
---

	memory B cells) and nasopharyngeal carriage of pneumococci (incl. antimicrobial resistance profiles) and H. influenza. Such a study is very welcomed, because randomized trials comparing vaccines from different producers head-to-head are rarely done. The manuscript is well written and comprehensive. I find the role of the reviewer for this type of paper (study protocol) difficult, because the content of this approved protocol is not open for discussion. Nevertheless, I have a few remarks/questions, to which you may consider providing some answers in the paper (p=page): p10, para 1: Please provide literature references to the trials mentioned in the last two sentences (POET, COMPAS, trials in Kenya, Finland, the Netherlands, Czech Republic). p10: The second paragraph and the first paragraph of section 6b could be moved to section 8 - Trial Design. There, also the co-administration of Infanrix hexa® should be mentioned. p11: In the last sentence, “immunogenicity of the co-administered vaccines” is mentioned as a primary (!?) objective, but I couldn’t find any information on how it will be assessed - should be mentioned in the Methods section. p12: I am surprised to read that “the primary objective is to compare a PCV13 schedule at 2, 4 and 9 months of age with a PCV10 schedule at 2, 3, 4 and 9 months of age.” If you find differences in immunogenicity between these two trial arms, how will you disentangle whether they are due to the different vaccines (PCV13 vs. PCV10) or due to the different schedules (2+1 vs. 3+1)? Why did you not choose the comparison of a 2+1 schedule with PCV13 vs. PCV10 as the PRIMARY objective? p14/15: In section 11, there is no sub-section marked as ‘11a’ - the first one is ‘11b’. p19: Last paragraph: The statistical power to detect differences in carriage rates is rather low. Will all trial participants be included in the NP sampling? p20: Recruitment: As recruitment is probably completed already, could you indicate the period of recruitment (and also the expected time line for follow-up and data analysis)? p23, para 1: Why did you choose the ITT population as your primary analysis? Since the objective is to compare different vaccination schedules, it would seem more plausible to me to base the PRIMARY statistical analysis on the per-protocol population.
--	--

VERSION 1 – AUTHOR RESPONSE

Reviewer 1

Comment 1. The study looks extensively at different schedules of PCV10, but only one of PCV13, and the reasons for this are not fully addressed.

Response: We used only one vaccine to evaluate the different vaccination schedules, as this permitted the inclusion of more schedules within the sample size/funding constraints. PCV10 was selected for practical reasons, in that the doses of PCV10 were donated whereas the doses of PCV13 had to be purchased at market price. A 2+1 PCV13 schedule was chosen for the head-to-head comparison with PCV10 as this was considered the most likely schedule to be introduced into Vietnam and other countries in the region.

Comment 2. The authors will be aware that the UK JCVI has recently announced that it will be recommending a switch to a 3, 12 month schedule for PCV13 based on the results of the Sched 3 study...it would be appropriate to reference these in this publication.

Response: Reference to the UK study comparing 1+1 and 2+1 schedules has been added to the Background and rationale (page 7)

Comment 3. It is notable that study participants were only recruited from areas within Ho Chi Minh City, and not in any regional areas...this is a potential weakness of the study design that could be discussed.

Response: We agree that findings may vary by epidemiological setting. However, for a Phase II study with laboratory-based endpoints, it was not practical to conduct this trial in rural areas.

Comment 4. The authors have said that one of the main limitations of the study is the fact that the nasopharyngeal data could be subject to natural variations in carriage, which could skew the data. However, all studies on nasopharyngeal carriage would be subject to this issue, and this is therefore not a limitation of this study specifically.

Response: The limitation in relation to potential variations in carriage is not so much about skewing the data, but that the study will have low power for the carriage endpoints if carriage rates are lower than anticipated. The bullet point for this limitation has been modified to clarify.

Comment 5. The description of the study groups and objectives in the abstract is quite awkward; these are complicated but some rewording here would be appropriate.

Response: The Abstract has been re-worded to improve readability.

Comment 6. There could also be a bit more precision in the language...

Response: The language has been modified to improve the precision as requested (in the Background and rationale on page 6 and in the Objectives on pages 9-10). Please note that the primary endpoint is also specified immediately after the two study questions are first mentioned on page 9.

Reviewer 2

Comment 1. Please consider changing the control group label "No intervention F", to "Experimental F"

Response: The labels "No intervention F" and "No intervention G" have been changed to "Control F" and "Control G", in line with the rest of the manuscript.

Comment 2. OPA analyses: the proposed analyses (% indices ≥ 8 and GMTs) are standard OPA analyses and will facilitate comparison to results of other trials, but may not be biologically meaningful...additional analyses that describe the distribution may be more informative to answer the research questions, such as comparing the highest value that 80% of the subjects have achieved.

Response: We will use the standard analyses as the main outcome, as these were pre-specified in the protocol and will facilitate comparison with other studies. However, we are aware of alternative approaches for describing OPA results, and will explore these (for example, reverse cumulative distribution plots) as additional post-hoc analyses alongside the standard analyses.

Comment 3. Background: a) the protocol states "there have been no published studies to date directly comparing [PCV10 and PCV13] post-primary series immunogenicity or impact on nasopharyngeal (NP) carriage." However, there have been three head-to-head trials of 3+0 or 2+1 schedules evaluating immunogenicity post-primary and two head-to-head NP carriage trials... b) ...recent [PRIME] systematic review...

Response: The studies identified are either not yet published (i, ii, v and vi) or describe post-booster rather than post-primary series immunogenicity (iii and iv), therefore the sentence "there have been no published studies to date directly comparing [PCV10 and PCV13] post-primary series immunogenicity or impact on NP carriage" remains true. Similarly, the results of the PRIME systematic review and meta-analyses are not yet available in the primary literature.

Comment 4. a) There is some lack of clarity re: whether or not the co-administered vaccines (measles and Infanrix-hexa) are being administered as per the routine EPI schedule for these children or whether one or more doses is being added... b) There is also some lack of clarity re: which concomitant vaccines will be given at 9 months to Group A (3+1 PCV) and how and why this differs from Groups C and E (2+1 PCV)... c) Specify the timing (ages) of the 4 Infanrix-hexa doses...

Response: The provision of Infanrix-hexa in the study has been clarified as follows: additional detail has been added to the Revision chronology section (page 3); provision of Infanrix-hexa has been added to the Trial Design section (page 11); and the Relevant concomitant care section (page 14) has been modified.

Comment 5. Are both pre-PCV blood draws at 18m and 19m necessary in children in Group G (who get 1 dose of PCV at 24m)?

Response: The main reason for the 19m blood in Group G is to look at the response to the 18m dose of Infanrix-hexa, comparing children who received Infanrix-hexa and children who received Quinvaxem in infancy. This objective has been added as a new section: Additional objectives (page 11).

Comment 6. Timeline: a) please add the concomitant vaccines being administered as part of the study... b) There are two blood draws in each group for which "each participant provides only one of these blood samples". Please describe the assignment process wrt when and how assigned...

Response: a) The concomitant vaccines have not been included in the Participant timeline so as to ensure that the table is clear to the reader (as these vaccines are given at different times in the different groups their inclusion would substantially increase the size of the table). However, the administration times for these vaccines have been added to the Relevant concomitant care section (page 14). b) Footnote 2 has been expanded to include detail of the assignment process for the timing of the blood samples.

Comment 7. Sample size: a) Only tests with 1-sided type I error are described. Please justify why the product comparison is not a test with a 2-sided type I error. b) Why is carriage assessed as a difference while immunogenicity is assessed as noninferiority? c) The difference in carriage between groups appears to have the same relative risk (0.6), regardless of which two groups are being compared... Please justify this assumption.

Response: a) Tests with one-sided type I error are described where we are interested in a difference in only one direction - this is standard for non-inferiority comparisons (MJA 2009;190(6):326-330), and is also true for the main carriage outcomes, where we are looking specifically for a reduction in carriage (firstly comparing 'fully-vaccinated' children with controls, and secondly to look for a reduction in carriage with a booster dose). For other comparisons in this study, two-sided type I error is used. b) The primary outcome of immunogenicity was designed to assess non-inferiority, in line with what is used by regulatory authorities to demonstrate that new schedules are not inferior to approved schedules. These regulatory expectations are not as relevant for the carriage outcomes that do not form part of regulatory approval, so we used the standard approach of tests of difference. c) There was no local data available to suggest how a booster dose might affect the anticipated carriage rates; therefore we used the same assumptions for the power calculation for the 3+1 vs 3+0 comparison as for the 3+1 vs controls comparison. We recognise that the power for these secondary outcomes is relatively low, and this is identified as a limitation of the study.

Comment 8. Recruitment: this section is supposed to describe “Strategies for achieving adequate participant enrolment to reach target sample size”. Please describe elements that convey confidence in the ability of the trial to meet the sample size...

Response: More detail has been added to the Recruitment section (page 19) to better show the ability of the trial to meet the sample size.

Comment 9. Are the CRFs available for public viewing and if so where?

Response: The CRFs are not available for public viewing.

Comment 10. Definition of analysis population relating to protocol non-adherence: the protocol states that the [primary] analysis will be intent-to-treat, which biases findings towards similarity (i.e., non-inferiority)... Please justify the choice for an ITT analysis over PP...

Response: We have planned to do both ITT and per-protocol analyses, and to present the results for both. We thank you for pointing out that the per-protocol analysis should be considered as the primary analysis, and the Populations of analysis section (page 22) has been modified accordingly (this detail was not part of the protocol submitted to the ethics committees).

Comment 11. Appendix 1 (Control information sheet): It says “Three blood tests will be taken... The blood tests are to check the response to the vaccines”. However, in this group PCV is given at the time of the last blood draw and therefore will not be used to evaluate the response to PCV and there is no mention of testing responses to Infanrix-hexa, Measles or Rubella. Please justify the inclusion of this sentence in this consent form.

Response: The blood tests in Control Group G are used to test the responses to Infanrix-hexa (this has been added to the Objectives section on page 11) and for comparison with Group F responses to PCV10.

Other minor items:

Comment 12. Abstract: a) suggest changing “one of six PCV schedules” to “one of six PCV infant schedules”... b) suggest changing “unvaccinated controls that receive PCV10 and 18 and 24 months” (since that sounds contradictory) to “controls that receive PCV10 and 18 and 24 months” c) add as secondary outcome measures, response to serotypes that are in PCV13 but not PCV10.

Response: a) and b) changes made (page 2); c) the description of secondary outcome measures has been removed to improve the clarity of the Abstract (see Reviewer 1, comment 5).

Comment 13. Background: a) “Three PCV schedules are currently in routine use around the world”... clarify this sentence to reflect that there are 3 schedules used for introduction of PCV to distinguish from maintenance schedules that have fewer doses. b) ... providing additional clarity would be useful regarding the statement “In developing countries, a 2+1 schedule with an earlier booster dose may be advantageous”... c) There is a more up-to-date and more relevant publication regarding the data of a single dose following a booster compared to 2+1 to add to the current text... d) The background states “Little is known about the effect of different PCV schedules on carriage”...; however, a recent systematic review... found no compelling evidence that one schedule consistently performed better over the other.

Response: a) b) and c) changes made (pages 6, 7 and 7, respectively); d) "Little is known about..." changed to "There have been few trials that evaluate..." to improve clarity. As noted in point 3, the results of the PRIME review are not yet available in the primary literature.

Comment 14. Trial registration - data set: a) provide phone number and mailing address of the contact person; b) provide title of the contact of scientific queries; c) Suggest changing the health condition studied from “pneumococcal vaccination” to “immune response to pneumococcal

vaccination”; d) item #21 (ethics review process information) – please respond – I believe this is pending at this stage? e) Although items #22-24 pertain to results..., which is not applicable to this study at this stage, planned or anticipated responses would be helpful and informative in the event this document is not updated

Response: a) and b) Title, name and email address of contact included; c) The health condition has been kept as "pneumococcal vaccination" as this trial includes NP carriage outcomes in addition to the immune responses; d) this section has been written based on the example provided in the SPIRIT 2013 explanation and elaboration: guidance for protocols of clinical trials (BMJ 2013;346:e7586), but Item 21 (ethics review process information) has been added as requested. The ClinicalTrials.gov record will be updated as required, therefore items 22-24 are not included here.

Comment 15. Objectives: a) It is more clear to specify ‘geometric mean IgG’ instead of just ‘IgG levels’; b) it is more clear to add “as measured by the proportion with protective levels of antibody”

Response: a) and b) changes to the wordings have been made (pages 9 and 10, respectively)

Comment 16. Interventions: specify whether PCV will be given in a different limb from concomitant vaccines

Response: This detail has been added to the Relevant concomitant care section (page 14)

Comment 17. Statistical methods: the protocol states that the booster response will be analyzed by ANCOVA – what are the planned covariates in this model that will be adjusted for?

Response: This detail has been added (page 21)

Comment 18. Statistical methods: the protocol states that “Further details of the planned statistical methods can be found in the Statistical Analysis Plan.” Please indicate where this can be found.

Response: This detail has been added (page 22)

Comment 19. Protocol amendments: please indicate whether or not there are plans to communicate protocol updates to the trial registry/journal.

Response: This detail has been added (page 24)

Comment 20. Access to data: access to the data: the protocol states that the ‘trial manager’ will have access to the full anonymized final dataset. Who is the trial manager and where are they located (i.e., what organization)? This statement implies no one else has access, including none of the study PIs, data analysts, statisticians – can you clarify?

Response: This statement has been expanded to show that the PI and trial statistician will also have access to the dataset.

Comment 21. Appendix 2: please describe where and how specimens be kept for long-term storage for future potential analyses/studies.

Response: The statement on specimen storage (paragraph 1) has been modified to clarify that this refers to long-term storage.

Reviewer 3

Comment 1. Please provide literature references to the trials mentioned in the last two sentences...

Response: Reference to the review paper has been moved to the end of the paragraph, to make it clear that this reference covers all the trials mentioned (page 8).

Comment 2. The second paragraph and the first paragraph of section 6b could be moved to section 8 - Trial Design. There, also the co-administration of Infanrix hexa® should be mentioned.

Response: Details of the vaccination schedules and the co-administration of Infanrix-hexa have been added to the Trial Design section (page 11)

Comment 3. In the last sentence, “immunogenicity of the co-administered vaccines” is mentioned as a primary (!?) objective, but I couldn’t find any information on how it will be assessed - should be mentioned in the Methods section.

Response: Immunogenicity of the co-administered vaccines has been removed from these objectives. However, this is an additional objective for Arm G (see addition on page 11), and has therefore been added to the methods (Immunogenicity of Infanrix-hexa on page 16) and to the end of Appendix 2.

Comment 4. I am surprised to read that “the primary objective is to compare a PCV13 schedule at 2, 4 and 9 months of age with a PCV10 schedule at 2, 3, 4 and 9 months of age.” If you find differences in immunogenicity between these two trial arms, how will you disentangle whether they are due to the different vaccines (PCV13 vs. PCV10) or due to the different schedules (2+1 vs. 3+1)? Why did you not choose the comparison of a 2+1 schedule with PCV13 vs. PCV10 as the PRIMARY objective?

Response: At the time of designing the trial, 2+1 was not a recommended schedule for PCV10. The 3+1 schedule was therefore selected as the comparator for the primary endpoint, with the 2+1 vs 2+1 comparison noted as a key secondary objective. The results of both these comparisons will be reported alongside each other, and used to help interpret any differences that might be found between the 2+1 PCV13 and 3+1 PCV10 arms.

Comment 5. In section 11, there is no sub-section marked as ‘11a’ - the first one is ‘11b’.

Response: The numbering of the headings has been removed in the revised version (the detail of the numbering is included in the SPIRIT checklist), but the first three paragraphs in the Interventions section (page 13) are section 11a.

Comment 6. The statistical power to detect differences in carriage rates is rather low. Will all trial participants be included in the NP sampling?

Response: Yes, all trial participants are included in the NP sampling. We recognise that the power for the carriage outcomes is rather low, and highlight the resulting vulnerability of these outcomes to fluctuations in carriage rates in the Limitations section on page 2. However, due to funding restrictions we were unable to increase the sample size for these secondary endpoints.

Comment 7. Recruitment: As recruitment is probably completed already, could you indicate the period of recruitment (and also the expected time line for follow-up and data analysis)?

Response: The number of enrolled participants and the periods of enrolment have been added to the Trial registration - data set on page 4.

Comment 8. Why did you choose the ITT population as your primary analysis? Since the objective is to compare different vaccination schedules, it would seem more plausible to me to base the PRIMARY statistical analysis on the per-protocol population.

Response: Please see response to Reviewer 2, comment 10.

VERSION 2 – REVIEW

REVIEWER	Gerhard Falkenhorst Robert Koch Institute, Department of Infectious Disease Epidemiology, Berlin, Germany
REVIEW RETURNED	29-Jan-2018

GENERAL COMMENTS	Dear authors, thank you for the revised version of the manuscript, in which you have adequately addressed my comments and suggestions - and those of the other reviewers, as far as I can tell. My recommendation to the editors is to accept the manuscript as is.
---

REVIEWER	Matthew Snape University of Oxford, United Kingdom
REVIEW RETURNED	11-Feb-2018

GENERAL COMMENTS	Thank you for addressing the concerns raised in my first round of reviews; these have been adequately addressed.
--

REVIEWER	Maria Deloria Knoll Johns Hopkins Bloomberg School of Public Health USA
REVIEW RETURNED	21-Feb-2018

GENERAL COMMENTS	Comments on Author's Response to Reviewer #1 I thank the authors for their sincere efforts in responding to the many comments that were made. They have addressed most of the issues and so listed here are just a few remaining points for clarification. I have specified when a response is desired. Re: comment 6b: This is still unclear. For example, the row for Group A has a footnote '2' for the 2m and the 18m blood draw and the footnote says 'each participant provides only one of these blood samples'. Therefore, I assume one child has a 2m blood draw but not an 18m blood draw and another child vice versa. The added clarification 'the last 50 participants per group enrolled into Groups A-E provide this blood sample at 18 months' explains who gets the 18m blood draw but not who gets the 2m blood draw. (Note that for Groups B & D the other blood draw is at 9m, for Group C it is at 6m, Group E 3m). Please clarify further in the footnote. Re: comment 7a: I interpret the response to mean that the product comparisons will be a 2-sided comparison. If my interpretation is incorrect, please make a correction; otherwise no further response is required. Re: comment 7b: (no response required) Note that whether or not to conduct a non-inferiority test depends on the question/hypothesis of interest, not whether or not the analysis is for an outcome that has been used as a basis for licensure or not. In this situation, a 1-sided null hypothesis could be that 3 doses is better than 2 doses and the alternative hypothesis is that 2 doses is not worse than 3 doses; while a 2-sided null hypothesis is that 2 doses = 3 doses and the alternative is that they are not equal. If the goal is to demonstrate "not worse than", then a 1-sided test is desired. This is because not finding a difference with a 2-sided test (e.g., $p > .05$) does not mean they are similar. Re: Comment 8: reaching the target sample size requires not just enrollment but also high completion rates (i.e., all immunization and blood draws collected in a timely manner). Please describe what will be done to ensure there are few missed/late immunization and blood draw visits. Re: comment 14C: I suggest changing the health condition from
--

	“pneumococcal vaccination” to “immunogenicity and NP carriage response to pneumococcal vaccination”. This is because ‘vaccination’ is not a health condition, it is an intervention. Health conditions pertain to the health state of the child, such as ‘sick’, ‘immune’, ‘colonized’, ‘died’, ‘recovered’ etc. Re: comment 18: The protocol states that “further details of the planned statistical methods are found in the Statistical Analysis Plan, located on a secure server at MCRI”. “Further details are found...” implies that we can find them. However, we do not have access to the server, obviously. So this sentence is confusing. If we have permission to see them, please include them in the appendix. If not, please clarify what is possible, such as ‘which can be provided upon request’ or some such statement. Otherwise perhaps delete this sentence.
--	--

VERSION 2 – AUTHOR RESPONSE

We thank the reviewers for considering the revisions made to the manuscript following the first round of comments. We note with thanks that the comments and concerns from Reviewer 1 and Reviewer 3 have been adequately addressed, with no further changes required. Please find below a point-by-point response to the remaining comments from Reviewer 2.

Re. comment 6b: This is still unclear... Please clarify further in the footnote.

Response: You are correct in your assumption that any participant that does not have the 18m blood draw has a blood draw at the other time point indicated by the footnote '2'. The footnote has been expanded to specify this.

Re. comment 7a: I interpret the response to mean that the product comparisons will be a 2-sided comparison.

Response: Some of the comparisons will be one-sided and some will be two-sided. One-sided comparisons will be used for the tests of non-inferiority in immunogenicity (2 vs 3 doses) and the tests of reduction in carriage (3+1 vs controls and 3+1 vs 3+0). Other comparisons will be two-sided; the details of which comparisons are one- or two-sided are specified in the Sample Size section (page 18).

Re. comment 7b: no response required.

Re. comment 8: reaching the target sample size requires not just enrollment but also high completion rates... Please describe what will be done to ensure there are few missed/late immunization and blood draw visits.

Response: According to the SPIRIT guidelines, and as noted in your original Comment 8, the Recruitment section (page 19) relates to strategies for achieving adequate participant enrolment. Information on participant retention is included in the Data collection methods section (page 20), and further detail has been added to this section in response to this comment.

Re. comment 14c: I suggest changing the health condition from "pneumococcal vaccination"...

Response: the health condition has been changed to "pneumococcal vaccination responses".

Re. comment 18: The protocol states that "further details of the planned statistical methods are found in the Statistical Analysis Plan, located on a secure server at MCRI"... please clarify what is possible... Otherwise perhaps delete this sentence.

Response: this sentence has been deleted.